# Bioengineered Mesenchymal-Stromal-Cell-Derived Extracellular Vesicles as an Improved Drug Delivery System: Methods and Applications

**DOI:** 10.3390/biomedicines11041231

**Published:** 2023-04-21

**Authors:** Cristiana Ulpiano, Cláudia L. da Silva, Gabriel A. Monteiro

**Affiliations:** 1Department of Bioengineering and iBB—Institute for Bioengineering and Biosciences, Instituto Superior Técnico, Universidade de Lisboa, Av. Rovisco Pais, 1049-001 Lisbon, Portugal; 2Associate Laboratory i4HB—Institute for Health and Bioeconomy at Instituto Superior Técnico, Universidade de Lisboa, Av. Rovisco Pais, 1049-001 Lisbon, Portugal

**Keywords:** mesenchymal stromal cells, extracellular vesicles, bioengineering strategies, drug delivery

## Abstract

Extracellular vesicles (EVs) are cell-derived nano-sized lipid membranous structures that modulate cell–cell communication by transporting a variety of biologically active cellular components. The potential of EVs in delivering functional cargos to targeted cells, their capacity to cross biological barriers, as well as their high modification flexibility, make them promising drug delivery vehicles for cell-free therapies. Mesenchymal stromal cells (MSCs) are known for their great paracrine trophic activity, which is largely sustained by the secretion of EVs. MSC-derived EVs (MSC-EVs) retain important features of the parental cells and can be bioengineered to improve their therapeutic payload and target specificity, demonstrating increased therapeutic potential in numerous pre-clinical animal models, including in the treatment of cancer and several degenerative diseases. Here, we review the fundamentals of EV biology and the bioengineering strategies currently available to maximize the therapeutic value of EVs, focusing on their cargo and surface manipulation. Then, a comprehensive overview of the methods and applications of bioengineered MSC-EVs is presented, while discussing the technical hurdles yet to be addressed before their clinical translation as therapeutic agents.

## 1. Introduction

Extracellular vesicles (EVs) are nano-sized lipid bilayer structures that enclose a variety of cellular components and mediators and facilitate the targeted delivery of their functional cargo to nearby or distant cells [1,2]. EVs have the intrinsic capacity to cross biological barriers, including plasma/endosomal membranes and the blood–brain barrier (BBB) [3], and demonstrate reduced immunogenicity and low toxicity in the spleen and liver [2,4]. These unique attributes are rendering EVs attractive drug delivery vehicles, allowing them to overcome limitations often associated with synthetic nanocarriers. In fact, EVs seem to be internalized more efficiently and deliver their therapeutic agent several orders of magnitude more efficiently than synthetic nanoparticles [5,6]. EVs can be further bioengineered to harbour exogenous cargoes or to alter surface properties improving their therapeutic efficacy and target-specificity [7,8].

Mesenchymal stromal cells (MSCs) are one of the most extensively explored cell types for cell-based therapeutics to treat a wide range of diseases. The main therapeutic attribute of MSCs is their ability to locally modulate the tissue microenvironment by secretion of a wide spectrum of trophic factors, including growth factors, cytokines and adhesion molecules [9,10]. These biologically active molecules can have a positive paracrine effect on tissue repair and regeneration, namely immunomodulation [11], inhibition of inflammation and anti-fibrosis [10,12], angiogenesis [13], support of proliferation and differentiation of progenitor cells and recruitment of endogenous cells [14,15]. For instance, MSCs significantly support wound healing by polarizing macrophage anti-inflammatory M2 activation, promoting angiogenesis and enhancing the survival and migration of fibroblasts [16]. In the context of central nervous system (CNS) regeneration, MSCs can facilitate neurogenesis by preventing apoptosis of endogenous neural cells and promoting axon re-extension by inhibiting the effect of extrinsic factors derived from the external environment of damaged areas [17].

EVs are among the major signalling effectors of the secretome of MSCs [18,19]. Since EVs are anticipated to preserve significant features of parental cells, MSCs have been extensively investigated as EV producers. Mesenchymal-stromal-cells-derived extracellular vesicles (MSC-EVs) share immunosuppressive activity and immunomodulatory properties with MSCs [20]. MSC-EVs are able to influence recipient cells, both at genetic and biochemical levels, exerting an additional regulatory effect by modulation of several physiological processes [18,19]. MSCs can be isolated from virtually all adult and fetal organs tested, including bone marrow (BM), adipose tissue (AT), Wharton’s jelly (WJ) and dental pulp (DP) [21,22], and have proven to be exceptionally safe, being presently approved for clinical use [21,22,23]. MSC-EVs can be considered safer than MSCs since these do not self-replicate and do not cause microvasculature entrapment [23,24]. Additionally, MSCs can efficiently mass-produce EVs by withstanding large-scale expansion and immortalization, enabling a sustainable and reproducible EV production process [25]. Lastly, MSC-EVs present good stability during storage, making them promising candidates for off-the-shelf therapeutics [26]. Altogether, these features suggest MSCs are a suitable candidate for the mass production of customized EVs.

This review paper will describe the basics of EVs biogenesis, composition and uptake, and the emerging strategies for bioengineering EVs with the aim of maximizing their therapeutic efficacy, either by cargo customization or surface functionalization. Based on this knowledge, a comprehensive overview of current methods and applications of bioengineered MSC-EVs will be presented. Finally, we discuss the advances and technical challenges yet to be addressed in the clinical translation of bioengineered MSC-EVs as standard therapeutic agents.

## 2. Fundamentals of EVs: Biogenesis, Composition and Uptake

Vesicle formation and secretion were first acknowledged in the 1980s by Johnstone and Stahl groups when investigating membrane biochemistry and trafficking during reticulocyte maturation [27,28], being identified as a cellular process for waste disposal [29]. Currently, it is strongly established that EVs are essential mediators of intercellular communication by transporting numerous proteins, lipids and nucleic acids, and thus able to modulate many normal physiological and pathological conditions (Figure 1a) [1,2]. Due to their robust potential as natural biomedicines, drug delivery systems (DDS) and diagnostic biomarkers, EVs have gained increasing attention in the past decade [30].

EVs are a heterogeneous population that is generally categorized into three subsets based on their biogenesis: exosomes, microvesicles and apoptotic bodies (Figure 1b) [1,2].

Exosomes (Exo) are small membrane vesicles with a diameter of 40 to 150 nm released from cells by the fusion of an intermediate organelle of the endocytic pathway—the multivesicular body (MVB)—with the cell surface. The biogenesis of Exo initiates with the formation of the MVB through the maturation of early endosomes. During this process, the membrane of MVBs suffers inward budding, forming intraluminal vesicles (ILVs) sequestering proteins and nucleic acids [1,2,7] that are specifically sorted by the endosomal sorting complex required for transport (ESCRT) [31], lipids (e.g., ceramides) [32], and tetraspanins [1,33]. MVB can either direct proteins to lysosomes for degradation or be transported and fused to the plasma membrane for the release of ILVs that are then referred to as Exo [1,2,7]. Due to their biogenesis, ESCRT proteins and their accessory proteins, such as ALG-2-interacting protein X (ALIX) and tumour susceptibility gene 101 (TSG101), are expected to be found in Exo regardless of the type of cell from which they originate. Other proteins reported to be abundant in Exo include membrane proteins of the tetraspanin family (e.g., CD63, CD9 and CD81), lysosome-associated membrane proteins (Lamps), heat shock proteins (HSP) and other cytosolic proteins, such as RAB GTPases and annexin that participate in intracellular trafficking [1,2,7].

Microvesicles (MVs) or ectosomes are larger membrane vesicles, ranging from 50 nm to 1 µm in diameter, that result from direct outward budding and fission of the plasma membrane [1,2]. This process is mediated by the local redistribution of the protein and lipid components of the plasma membrane, which modulates changes in membrane curvature and rigidity [34]. Ca^2+^ accumulation induces the activation of proteolytic enzymes (e.g., calpain) and lipid translocases (e.g., flippases, floppases and scramblases) that disrupt the equilibrium of the phospholipids between the two leaflets that cause the physical bending of the membrane and loss of membrane–cytoskeleton connection, facilitating vesicular release [1,35]. Additionally, membrane budding is associated with lipid rafts which are specialized regions of the plasma membrane that are enriched in cholesterol, glycosphingolipids and glycosyl-phosphatidylinositol (GPI)-anchored proteins. Caveolin-1, a structural protein of caveolae lipid rafts, has been shown to regulate the formation and cargo sorting of MV [36]. Although membrane budding occurs through a different process than Exo formation, it also depends on endosomal machinery, including the ESCRT components, tetraspanins and RAS GTPases [1,34]. Other proteins that are found abundant in MVs include cytoskeletal proteins, such as actin, and plasma-membrane-associated proteins [1,35,36].

Finally, apoptotic bodies (ApoBDs) that range from 500 nm to 2 µm in diameter are also generated from the cell surface, although these are only released during the disassembly of an apoptotic cell into subcellular fragments. As a result, ApoBDs contain a wide range of cellular components, possibly including chromatin/DNA fragments, cytosol portions, degraded proteins or even intact organelles [37].

Besides the sorted proteins, Exo and MVs also contain a variety of nucleic acids, including DNA, messenger RNA (mRNA) and different classes of non-coding RNAs, namely micro RNAs (miR), long non-coding RNAs and circular RNAs [1,2,7]. Although the exact mechanism that regulates the sorting of RNA species into EVs is still unknown, some RNA-binding proteins (RBP) have been found to participate in RNA sorting through the recognition of specific sequence motifs. For example, SUMOylated heterogeneous nuclear ribonucleoprotein A2/B1 is an RBP and has been reported to regulate miR trafficking into EVs by binding to specific motifs (GGAG/CCCU) [38].

Once secreted, EVs can interact with the target cells either located within the microenvironment or in distant sites travelling through blood and other body fluids. This interaction is facilitated by numerous mediators, including tetraspanins, integrins, lipids, lectins, heparan sulfate proteoglycans and other extracellular matrix (ECM) components [1]. The direct binding of EVs can induce a downstream signalling cascade in the recipient cell via ligand–receptor interactions (e.g., antigen presentation, immune modulation and morphogen signalling) [1,7]. For instance, EVs have been reported to act as carriers in the long-range transfer of the canonical lipid-anchored morphogens Hedgehog (Hh) and Wnts to recipient cells which induce several physiological processes, such as stem cell maintenance, tissue repair and metabolism [39].

Alternatively, EVs can transfer their intraluminal cargo to the recipient cells either by direct membrane fusion or endocytosis. Endocytosis is the main uptake mechanism and occurs through different pathways: receptor-mediated endocytosis, clathrin-mediated endocytosis, caveolin-mediated endocytosis, lipid raft-mediated endocytosis, phagocytosis and micropinocytosis [1,7]. The internalized EVs follow the early endosomal pathway in which they can either be recycled back to the plasma membrane, degraded in the lysosome and used as a metabolites source, or undergo endosomal escape, through back fusion with the limiting membrane MVB, releasing their contents to the cytosol (Figure 1c) [1,7].

Although the mechanism by which cells discriminate the fate of the internalized EVs is poorly understood, the delivery capacity of EVs has been widely demonstrated. The release of their intraluminal content triggers alterations in the recipient cells by the action of nucleic acids, including miRNA and mRNA, that regulate gene expression, and other important genetic elements, including genomic DNAs, mitochondrial DNAs and long noncoding RNAs [1,7]. EVs also release protein and peptide cargos that induce a functional response in the recipient cells. For example, in dendritic cells, protein cargos of EVs can be processed and used in antigen presentation regulating immune response [1,40].

Due to the overlapping sizes and absence of proteins that are restricted to each population, in this review, all the different vesicles will be collectively referred to as EVs, as proposed by the International Society for Extracellular Vesicles (ISEV) [40].

## 3. Strategies to Maximize the Therapeutic Efficacy of EVs

Despite the intrinsic potential of EVs as natural delivery vehicles, bioengineering techniques have been applied to maximize their therapeutic efficacy. This can be achieved using two major strategies: cargo engineering and surface engineering. Essentially, by customizing the therapeutic payload of EVs or enhancing their selectivity to target cells, bioengineered EVs have the potential to become more personalized and targeted therapeutics (Figure 2).

### 3.1. Cargo Customization

EVs are being explored as natural nanocarriers through their artificial loading with different therapeutic agents, including small molecules, drugs, proteins and different RNA species, such as small interference RNA (siR) and miR. The incorporation of extrinsic cargo into EVs requires the manipulation of the EVs or the parental cells. This can be accomplished by two methods: exogenous/direct loading, with the external incorporation of cargo into isolated EVs, and endogenous/indirect loading, by providing the parental cells with the means to naturally incorporate the desired cargo during EV biogenesis (Figure 2a).

#### 3.1.1. Exogenous Cargo Loading

Exogenous loading occurs after EV isolation by direct encapsulation of the desired therapeutic cargo through various processes, including co-incubation [41,42,43,44], electroporation [44,45,46,47,48], sonication [43,44,49,50], freeze-thawing [43], extrusion [50,51] and permeation by a detergent-based compound [43,50] (Figure 2a).

Incubation is a passive loading method that has been used to encapsulate hydrophobic drugs into EVs. For example, EV co-incubation with curcumin (Cur) improved its bioavailability and anti-inflammatory effect in a mouse model of inflammation [41]. Similarly, hydrophobically modified siR were successfully encapsulated in EVs derived from glioblastoma cells through co-incubation [42].

Alternatively, different active loading strategies have been employed to physically or chemically permeabilize the hydrophobic membrane of EVs, allowing the transient diffusion of hydrophilic molecules into their intraluminal space.

For instance, electroporation relies on the exposure of the EV membrane to high-intensity electrical pulses and has been widely used to facilitate the loading of different cargos, including siR [3,45], miR [46,52,53], DNA [54,55] and other small molecules [47]. Moreover, Usman and collaborators explored electroporation to engineer EVs derived from human red blood cells (RBC) for the delivery of antisense oligonucleotides, Cas9 mRNA and guide RNAs, to CRISPR–Cas9 edit the recipient cells [48].

The sonication method has also been described to promote the active loading of a variety of small nucleic acids into EVs, using a low-intensity ultrasound frequency [49]. Interestingly, Kim and colleagues reported that sonication provided the greatest loading capacity of paclitaxel (PTX) into macrophage-derived EVs when compared to incubation and electroporation [44].

Freeze-thawing involves the combination of EVs with the cargo at room temperature, followed by repeated cycles of freezing (at −80 °C or in liquid nitrogen) and thawing, allowing for cargo incorporation through membrane deformation. Although some studies reported that freeze-thawing led to EV aggregation [43,56], Hettich and co-workers showed that this method demonstrated a great loading efficiency of hydrophilic compounds while maintaining the structural and biological characteristics of the EVs [57]. This method has also been applied to produce hybrid vesicles by actively fusing the membrane of EVs and liposomes (further reviewed below) [56].

Alternatively, permeation by the detergent-based compound saponin is a method used to chemically load EVs, which induces the formation of membrane pores without its destruction by removing cholesterol. For instance, this technique was applied to load the large protein catalase into EVs derived from macrophages, which resulted in a loading efficiency comparable to the sonication and extrusion methods [50]. This method showed the most efficient loading (~50%) when encapsulating doxorubicin (DOX), compared to 37 °C and room temperature co-incubations and freeze–thaw cycles [43].

Finally, extrusion is a technique used to artificially produce vesicles by breaking up the cells and then reforming the contents into EV mimetics while retaining some of the physical and biological characteristics of secreted EVs. For example, Jang and colleagues produced EV mimetics from monocytes or macrophages harbouring different chemotherapeutic drugs using serial extrusion through filters with diminishing pore sizes (10, 5 and 1 μm). Remarkably, EV mimetics presented a similar in vivo anti-tumour activity compared to naturally secreted EVs [51].

Overall, the incubation method is a straightforward strategy that preserves the integrity of the EV membrane, but it has low loading efficiency and is only compatible with hydrophobic cargos. In opposition, active loading methods present higher loading efficiencies. However, these are still limited in their technical complexity and often disrupt membrane/cargo integrity and stability and promote aggregation [43,44,50,58,59,60].

#### 3.1.2. Endogenous Cargo Loading

Endogenous loading depends on the availability of desired cargo in the producer cell and the subsequent use of the cellular machinery for its incorporation into EVs. The introduction of the exogenous cargo into the producer cell can be achieved by passive loading through simple incubation or active loading through the genetic manipulation of the parental cells (Figure 2a). Simple incubation has been mostly used to endogenous incorporate small drugs into EVs [61,62].

By contrast, genetic manipulation has been used to bioengineer EVs to harbour small non-coding RNAs and mRNA/proteins of interest. For instance, THP-1 monocytes genetically engineered to transiently overexpress miR-939 secreted EVs loaded with the produced miR. One possible cellular mechanism for its incorporation into EVs was the recognition of its RBP binding motif GGAG [52]. Endogenous miR loading was also studied by Lee and colleagues through the engineering of a stable producer cell line. Human embryonic kidney (HEK) 293T cells were engineered to express the miR-124, typically repressed in Huntington’s disease (HD), using a retroviral expression system. The produced EVs were enriched in miR-124 and induced the silencing of the target gene REST after injections into a mouse HD model [63]. Active and specific RNA loading into EVs can be improved using the targeted and modular EV loading (TAMEL) technology that resorts to an EV-enriched protein to anchor an RNA-binding domain (RBD) on the intraluminal side of EVs. Essentially, the RBD binds to RNA presenting the specific sequence motifs and actively incorporates them into EVs. Hung and colleagues constructed a plasmid encoding for EV-enriched protein Lamp2b fused with the RBD MS2. This approach substantially enhanced the loading of RNA cargo with the sequence recognized by the MS2-RBD (up to six-fold) [64]. Similarly, EVs were engineered to load a specific RNA by fusing the tetraspanin CD9 with human antigen R (HuR), an RBP that interacts with miR-155 with high affinity [65].

Apart from small RNAs, proteins and mRNAs can also be endogenously loaded into EVs by transfection/transduction of the parental cells with the gene encoding the desired cargo. Mizrak and co-workers first reported that overexpression of the desired gene prompts the loading of the corresponding mRNA and protein into EVs. The authors transfected HEK 293T cells to express high levels of the enzyme cytosine deaminase fused to uracil phosphoribosyltransferase (CD-UPRT) that converts the prodrug 5-fluorocytosine (5-FC) into a cytotoxic cancer agent 5-fluorouracil. The isolated EVs were loaded with CD-UPRT mRNA/protein, inducing in vivo tumour regression in a mouse model upon injection and systemic treatment with 5-FC [66]. Similarly, in another study, A549 lung cancer cells were transduced with an adenoviral vector encoding the cystic fibrosis transmembrane conductance regulator gene, which is mutated in patients with cystic fibrosis (CF). EV-mediated delivery of produced mRNA/protein to CF cells corrected the deficiency in chloride channel activity [67].

Synthetic therapeutics can also be encapsulated into EVs by transfection of the parental cells. For example, after transfection of HEK 293T cells with synthetic siR that targets the expression of hepatocyte growth factor (HGF), there was an increased secretion of HGF-harbouring EVs with an inhibitory effect on tumour growth and angiogenesis in vitro and in vivo [68].

Generally, endogenous loading strategies allow relatively simple and stable production of EVs with engineered cargo, while preserving EV membrane integrity and the function of the loaded cargo. However, these approaches are often time-consuming and expensive compared to exogenous methods, have limited loading efficiency and can have a negative impact on parental cells [59,60].

### 3.2. Surface Functionalization

In vivo-administered EVs suffer from rapid clearance mostly by uptake into cells in the liver, spleen, gastrointestinal tract and lungs [69]. The surface of EVs is critical for their biodistribution, tropism and therapeutic effect and its modification can endow EVs with additional targeting to specific cell types, abilities to cross different biological barriers and extended lifespan in vivo until reaching the target location [56,70,71]. Numerous strategies have been investigated to functionalize the surface of EVs: genetic manipulation, by engineering the parental cells to produce EVs displaying transmembrane targeting moieties; chemical modification, by anchoring targeting moieties to the surface of isolated EVs; and hybrid membrane engineering, by conjugating isolated natural EVs and synthetic liposome nanoparticles (Figure 2b).

#### 3.2.1. Genetic Manipulation of Parental Cells

Typically, the parental cells can be genetically engineered to produce EVs with the desired surface feature by modifying native EV transmembrane proteins with exogenous ligands that are recognized by the recipient target cells, including proteins/peptides, antibodies and lipid-raft-associated components (Figure 2b).

In a pioneering study by Alvarez-Erviti et al., the surface of dendritic-cell-derived EVs was engineered to improve their brain targeting after systemic administration. Targeting was achieved by transfection with a plasmid encoding Lamp2b fused to the central nervous system (CNS)-specific rabies viral glycoprotein (RVG), resulting in increased brain accumulation after intravenous injections in a mouse model [3]. Since then, Lamp2b has been the most widely used protein anchor in surface engineering approaches. For instance, Lamp2b fused to αγ integrin-specific peptide iRGD and human epidermal growth factor receptor 2 (HER2)-binding affibody, showed improved EV tropism towards integrin-positive breast cancer cells and HER2-expressing tumour cells, respectively [47,72].

Other transmembrane proteins are used to anchor specific ligands. Liang and collaborators engineered HEK 293T cells to express a fusion between tetraspanin CD63 and Apo-A, a known target of the scavenger receptor class B type 1 receptor that is highly expressed by liver cancer cells. The produced EVs were effectively internalized by human liver cancer HepG2 cells via receptor-mediated endocytosis [53]. Similarly, due to its localization on the membrane of EVs, the C1C2 domain of lactadherin has been explored as an anchor for different recombinant proteins, such as carcinoembryonic antigen and HER2 [73]. Moreover, GPI-anchored proteins (associated with lipid rafts) were used to display a nanobody that targets epidermal growth factor receptor (EGFR), a well-studied oncogene, on the surface of EVs to target tumour cells expressing this receptor [74].

Apart from improving the affinity and selectivity of EVs to target cells/tissues, genetic modification can be used to produce EVs displaying tags that increase their lifespan. Kamerkar and colleagues engineered fibroblasts to overexpress CD14, which is an integrin-associated transmembrane protein described to protect cells from phagocytosis. CD47-enriched EVs showed higher circulation retention times by evading phagocytosis by monocytes and macrophages, in a mouse model [71].

#### 3.2.2. Chemical Modification

Alternatively, the targeting ligands can be incorporated into the surface of EVs by chemical modification after their isolation, relying on covalent bonds, hydrophobic insertions, lipid self-assembly or other non-covalent reactions (Figure 2b).

The simplest method is by the direct incorporation of hydrophobic/amphiphilic molecules into the naturally hydrophobic membrane of EVs. Phospholipid 1,2-dimyristoyl-sn-glycero-3-phosphoethanolamine (DMPE) and 1,2-distearoyl-sn-glycero-3-phosphoethanolamine (DSPE) polyethylene glycol (PEG) derivatives can accumulate in the membrane of EVs and have been successfully used to immobilize targeting ligands. For instance, macrophage-derived EVs containing PTX were modified with anisamide–DSPE–PEG moiety to target the sigma receptor, which is overexpressed by lung cancer cells [75]. Additionally, phospholipid–PEG derivatives increase EV stability in vivo, prolonging circulation times that potentially increase the accumulation of EVs in target tissues and specific cargo delivery [76]. Similarly to phospholipid derivatives, cholesterol can self-assemble into EVs due to its hydrophobicity. In this context, Huang and colleagues explored the potential of cholesterol-conjugated AS1411 DNA aptamer to mediate the targeted delivery of EVs to nucleolin, which is overexpressed on the surface of leukaemia cells [77]. Likewise, a bacteriophage Φ29 RNA has been engineered to incorporate cholesterol-conjugated EGFR RNA aptamer and used to decorate EVs carrying siR as a targeted anti-tumour treatment [78].

Click chemistry (copper-catalysed azide–alkyne cycloaddition) is a highly efficient covalent reaction between an alkyne and azide that forms a triazole linkage, which has been successfully applied to functionalize the surface of EVs [79]. In a study by Lee et al., alkyne-functionalised EVs were decorated with various functional agents using copper-free click chemistry, to allow their specific delivery to cancer cells [80]. Moreover, Jia and colleagues conjugated the membrane of macrophage-derived EVs with RGERPPR peptide, a specific ligand of neuropilin-1 (NPR-1) which is overexpressed in glioma cells, using a cycloaddition reaction with sulfonyl azide. The peptide-displaying EVs were able to cross the BBB and facilitate glioma recognition [70].

Non-covalent approaches include receptor–ligand binding and electrostatic interactions and have also been implemented to functionalize the surface of EVs. For instance, superparamagnetic nanoparticles (NPs) were conjugated to the transferrin receptors of blood-derived EVs. This strategy allowed an efficient separation of EVs from the blood and endowed EVs with a robust targeting ability under an external magnetic field [81]. Relying on electrostatic interactions, Nakase and Futaki engineered HeLa-derived EVs with a combination of cationic lipids and a pH-sensitive fusogenic peptide GALA, enhancing cell membrane binding and EV uptake, and subsequent cytosolic release of their cargo [82].

#### 3.2.3. Hybrid Membrane Engineering

The surface of EVs can also be functionalized using hybrid membrane engineering that results from the capacity of the lipid bilayer of EVs to spontaneously fuse with other membrane structures. Isolated natural EVs and decorated synthetic liposomes can be fused into hybrid nanoparticles without affecting their intrinsic properties (Figure 2b).

The surface properties of EVs can be easily modified using liposomes embedded with peptides/antibodies as targeting moieties. In this context, Li and colleagues fused tumour cell-derived EVs with liposomes modified with tumour-targeting peptides. The hybrid EVs allowed highly efficient drug loading and were strongly enriched in the tumour areas [83]. Sato and collaborators formulated engineered hybrid EVs by fusing their membrane to different synthetic phospholipid liposomes using the freeze–thaw method and confirmed that the delivery function of the EVs can be modified by changing their properties and lipid composition [56]. Moreover, EV fusion with functionalized liposomes via PEG-mediated reaction has been shown to facilitate the enrichment of EVs with exogenous lipophilic or hydrophilic compounds, while preserving their intrinsic content and biological properties [84]. In addition, through simple incubation, EV–liposome nanovesicles were generated and demonstrated to be able to efficiently encapsulate large plasmids, including the CRISPR-Cas9 expression vectors, similar to the liposomes. However, these could be endocytosed by MSCs and express the encapsulated genes, unlike liposomes [85]. Using the co-extruding method, Jhan and co-workers fused EVs with a suspension of different synthetic lipids by serial extrusion through membranes (400, 200 and 100 nm). This method allowed the formation of vesicles with controlled size and a 43-fold increase in production compared to native EV secretion [86]. In an alternative approach by Zhang et al., hybrid membrane engineering was used to develop multifunctional artificial EVs. Essentially, membrane proteins from RBC and breast cancer cells were incorporated into synthetic liposomes. The engineered hybrid EVs exhibited an anti-phagocytic capacity during circulation (high level of CD47 from RBC) and a tumour-homing ability (EpCAM, galectin 3 and N-cadherin from cancer cells) for targeted drug delivery [87].

## 4. Current Applications of MSC-EVs

Extensive research has shown that unmodified MSC-EVs have various therapeutic roles including immune regulation, anti-inflammatory effects and tissue regeneration [88]. Nevertheless, improvements are still needed in what concerns their targeting and payload potency, and thus bioengineering strategies have been widely employed to potentiate the benefits of MSC-EVs. This section describes and summarises a selection of current methods and applications of bioengineered MSC-EVs.

### 4.1. Loading MSC-EVs with Therapeutic Cargo

Different types of therapeutic payloads have been loaded into MSC-EVs, including nucleic acids, proteins and small molecules. Either endogenously, by manipulation of parental MSC, or exogenously, through manipulation of isolated EVs, cargo engineering of MSC-EVs has been shown to improve their therapeutic efficacy in particular clinical applications (Table 1).

miRs are promising new therapeutics for treating many diseases. MSC-EVs have emerged as a promising vehicle for delivering miRs, thus many researchers have been engineering MSC to express or harbour these molecules.

For instance, human BM-derived MSC (MSC(M)) were transduced with lentivirus vectors containing miR-124a which silences Forkhead box A2 expression, inducing aberrant intracellular lipid accumulation. Quantitative PCR demonstrated that the produced EVs contained approximately 60-fold higher levels of miR-124a compared to non-modified MSC-derived EVs. miR-124a-carrying EVs resulted in a significant in vitro reduction in viability and clonogenicity of glioma stem cells (GSC), and treated mice harbouring intracranial GSC xenografts after systemic administration [89]. In addition, other miRs have been endogenously loaded into MSC-EVs and demonstrated anti-cancer potential, including miR-379 [90], miR-16-5p [91] and miR-424 [92] through post-transcriptional regulation of tumour-related gene expression of ciclo-oxigenase-2, integrin α2 and transcriptional factors MYB, respectively.

**Table 1 biomedicines-11-01231-t001:** Overview of the potential strategies and applications of cargo-engineered mesenchymal-stromal-cell-derived extracellular vesicles (MSC-EVs).

Type of Strategy	Cargo	Application	Therapeutic Effect	MSC Source	Ref.
Nucleic Acids					
Endogenous loading (transduction)	miR-122	Liver fibrosis	Inhibited fibrosis	Human/mouse AT	[93]
	miR-124a	Glioblastoma	Increased survival of GSC-injected mice	Human BM	[89]
	miR-126	Skin wounds	Increased re-epithelialization, angiogenesis, and collagen maturity	Human SM	[94]
	miR-17-92	Ischemic stroke	Enhanced axon-myelin remodelling and functional recovery after stroke	Rat BM	[95]
	miR-379	Breast cancer	Inhibited tumour growth	Human BM	[90]
	miR-let7c	Renal fibrosis	Decreased fibrosis	Human BM	[96]
	mRNA-CD-UPRT	Cancer	Inhibited tumour growth	Human AT, BM, DP and WJ	[97]
Endogenous loading (transfection)	miR-126	Ischemic stroke	Increased neurogenesis and improved functional recovery after stroke	Rat AT	[98]
	miR-133b	Spinal cord injury	Inhibited inflammatory response and induced nerve function repair	Rat BM	[99]
	miR-150-5p	Rheumatoid arthritis	Inhibited synoviocyte hyperplasia and angiogenesis	Mouse BM	[100]
	miR-155-5p	Osteoarthritis	Increased proliferation and migration, suppressed apoptosis and enhanced ECM secretion of osteoarthritic chondrocytes	Human SM	[101]
	miR-16-5p	Colorectal cancer	Inhibited tumour growth	Human BM	[91]
	miR-181c	Burn-induced inflammation	Decreased inflammation	Human WJ	[102]
	miR-22	Spinal cord injury	Inhibited inflammatory response and induced nerve function repair	Rat BM	[103]
	miR-26a	Spinal cord injury	Promoted axonal regeneration and neurogenesis	Rat BM	[104]
	miR-29b	Alzheimer’s disease	Reduced the pathological effects of amyloid-β peptides	Rat BM	[105]
	miR-424	Ovarian cancer	Inhibited tumorigenesis and angiogenesis	Human BM	[92]
	miR-92a-3p	Osteoarthritis	Enhanced cartilage development and homeostasis	Human BM	[106]
Exogenous loading (electroporation)	miR-124	Ischemic stroke	Increased neurogenesis	Mouse BM	[107]
	miR-132	Myocardial infarction	Enhanced neovascularization and preserved heart functions	Mouse BM	[108]
	miR-499a-5p	Endometrial cancer	Inhibited tumour growth and metastasis	Mouse BM	[109]
	miR-590-3p	Myocardial infarction	Promoted cardiomyocyte proliferation and cardiac regeneration	Rat BM	[110]
	siR-CTGF	Spinal cord injury	Increased axon regeneration and motor function after SCI	Rat BM	[111]
	siR-galectin-9	Pancreatic ductal adenocarcinoma	Inhibited tumour growth	Human BM	[112]
	siR-Kras	Pancreatic ductal adenocarcinoma	Inhibited tumour growth	Human BM	[113]
	siR-PLK-1	Bladder cancer	Increased cytotoxicity and apoptosis	Human BM	[114]
	si-SHN3	Osteoporosis	Enhanced osteogenic differentiation and vessel formation and inhibited osteoclast formation	iPSC	[115]
Exogenous loading (incubation)	cholesterol-modified miR-210	Ischemic stroke	Increased angiogenesis and survival of ischemic brain mice	Mouse BM	[116]
	siR-PTEN	Spinal cord injury	Increased functional recovery of spinal cord lesion in rats	Human BM	[117]
Exogenous loading (transfection reagent)	miR-326	Inflammatory bowel disease	Inhibited the synthesis and production of inflammatory factors	Human WJ	[118]
**Proteins**					
Endogenous loading (protein transduction)	Akt	Myocardial infarction	Increased angiogenesis and cardiac regeneration	Human WJ	[119]
	Ang-2	Skin wounds	Increased angiogenesis and accelerated wound healing	Human WJ	[120]
	Osteoactivin	Osteoporosis	Increased proliferation and osteogenesis of MSC and attenuated bone loss in ovariectomized rat	Rat BM	[121]
	PEDF	Ischemic stroke	Ameliorated cerebral ischemia–reperfusion injury in rats	Rat AT	[122]
**Small molecules**					
Endogenous loading (incubation)	Iron oxide NPs	Skin wounds	Improved targeting under an external magnetic field and enhanced wound healing	Human WJ	[123]
	PTX	Pancreatic adenocarcinoma	Decreased tumour growth	Mouse BM	[62]
	TXL	Metastatic breast cancer; ovarian cancer; lung carcinoma	Inhibited tumour growth	Human WJ	[124]
	Venofer	Cancer	Increased tumour cell death under an external magnetic field	Human AT, BM, DP and WJ	[125]
Exogenous loading (dialysis)	DOX	Osteosarcoma	Inhibited tumour growth	Mouse BM	[126]
Exogenous loading (electroporation)	DOX	Colon adenocarcinoma	Inhibited tumour growth	Mouse BM	[127]
	NCTD	Hepatocellular carcinoma	Inhibited tumour growth	Human BM	[128]
Exogenous loading (electroporation/sonication)	GEMP/PTX	Pancreatic ductal adenocarcinoma	Increased homing and penetration, and anti-tumour potency	Human BM	[129]
Exogenous loading (extrusion)	PTX	Breast cancer	Decreased tumour growth	Human BM	[130]
Exogenous loading (freeze–thaw)	polypyrrole NPs	Diabetic peripheral neuropathy	Reduced the neural and muscular damage under electric stimulation	Rat BM	[131]
Exogenous loading (incubation)	Cur	Ischemic stroke	Decreased inflammation	Mouse BM	[132]
Exogenous loading (incubation; sonication)	TKI	Anaplastic thyroid cancer	Increased radioiodine sensitivity	Human AT	[133]

AT—adipose tissue; BM—bone marrow; CD-UPRT—cytosine deaminase fused to uracil phosphoribosyltransferase; CTGF—connective tissue growth factor; Cur—curcumin; DOX—doxorubicin; DP—dental pulp; ECM—extracellular matrix; GEMP—gemcitabine monophosphate; GSC—glioma stem cells; iPSC—induced pluripotent stem cells; NCTD—norcantharidin; NPs—nanoparticles; PEDF—pigment epithelium-derived factor; PLK-1—serine/threonine-protein kinase; PTX—paclitaxel; SCI—spinal cord injury; SHN3—schnurri-3 protein; SM—synovial membrane; TKI—tyrosine kinase inhibitor; TXL—taxol; WJ—Wharton’s jelly.

In the context of skin regeneration, Li and colleagues explored the potential of EVs derived from MSCs transfected with miR-181, which has a critical role in regulating inflammation, specifically in attenuating skin-burn-induced inflammation. The results demonstrated that the engineered EVs suppressed the TLR4 signalling pathway, reducing NF-κB/p65 activation, and alleviated inflammation in burned rats more effectively than EVs produced by non-transfected MSC [102]. Furthermore, in a study by Tao et al., EVs secreted by MSCs derived from the synovial membrane (SM) engineered to overexpress miR-126 were demonstrated to be able to heal full-thickness skin defects in a diabetic rat model [94]. Interestingly, MSC(AT)-derived EVs endogenously loaded with miR-126 also showed prospective effects in the treatment of ischemic stroke [98].

MSCs have also been engineered to produce miR-containing EVs that attenuate fibrosis. In a study by Lou et al., miR-122-engineered EVs inhibited fibrosis by reducing proliferation and collagen maturation of hepatic stellate cells through miR-122-induced downregulation of target genes such as insulin-like growth factor receptor-1, cyclin G-1 and prolyl-4-hydroxylase α-1 [93]. Moreover, MSC(M) engineered to overexpress miR-let7c generated EVs that inhibited the upregulated expression of fibrotic genes in neighbouring rat kidney tubular epithelial cells and attenuated renal fibrosis in vivo in a mouse model of unilateral ureteral obstruction [96].

Envisioning the treatment of rheumatoid arthritis (RA), MSC-EVs were engineered to harbour miR-150-5p, by transfection of the parental cells. miR-150-5p-loaded EVs decreased the migration and invasion of RA synoviocytes and downregulated tube formation in vitro, by targeting matrix metalloproteinase 14 and vascular endothelial growth factor. These MSC-EVs also reduced clinical arthritic scores and joint destruction in an in vivo RA mouse model [100]. In addition, a study on EVs derived from miR-92a-3p-expressing MSCs showed enhanced cartilage development and prevented its degradation by targeting wnt5a in a collagenase-induced osteoarthritis (OA) mouse model [106]. Furthermore, EVs secreted by miR-155-5p-overexpressing SM-derived MSCs promoted ECM secretion in vitro by targeting Runx2 and effectively prevented OA in a mouse model [101].

Studies have found that MSC-EV-mediated delivery of miR showed a positive effect in neurodegenerative diseases and mitigated the damage caused by CNS injuries. Jahangard and colleagues engineered MSCs to produce EVs encapsulating miR-29, which is downregulated in Alzheimer’s disease (AD) and silences the expression β-site amyloid precursor protein cleaving enzyme 1 and Bcl-2 interacting mediator of cell death. miR-29-EVs caused a reduction in the pathological effects of amyloid-β peptides after injection into the hippocampus of a rat model of AD, namely by improving spatial learning and memory deficits [105]. Furthermore, the lentivirus-based modification of MSCs to overexpress miR-17-92 allowed the production of EVs that enhanced axon–myelin remodelling and motor electrophysiological recovery after stroke in an in vivo mouse model [95]. Spinal cord injury (SCI) recovery has been investigated using MSC-EVs endogenously loaded with miR-133b, which is a key player in the differentiation of neurons and the outgrowth of neurites. miR-133b-EVs have been shown to activate signalling pathway proteins involved in the survival of neurons and the regeneration of axons, reduce the volume of the lesion and promote the regeneration of axons after systemic injection into a rat model of SCI [99]. In addition, recent studies reported that MSC-EV-mediated delivery of miR-22 and miR-26a could represent novel therapeutic approaches for the treatment of SCI [103,104].

To date, only a few studies have reported developing mRNA-loaded EVs. Among these, a study by Altanerova et al. describes a strategy where MSCs from different tissue sources were modified by retrovirus transduction to overexpress the suicide gene CD-UPRT. The mRNA-CD-UPRT was incorporated into the secreted EVs and induced cell death in the presence of prodrug 5-FC upon internalization by tumour cells [97].

Therapeutic siRs have also been delivered using MSC-EVs. For example, MSC(M)-derived EVs were electroporated with siRs targeting oncogenic Kras. The modified MSC-EVs induced the suppression of oncogenic Kras and increased the survival of several mouse models with pancreatic cancer [113]. Similarly, serine/threonine protein kinase (PLK-1)-targeting siRs were electroporated into MSC(M)-derived EVs. siR-PLK-1-carrying EV delivery to bladder cancer cells resulted in the suppression of PLK-1 and contributed to cell cycle arrest and apoptosis [114]. In a different context, MSC-EVs were loaded with a siR that silences the expression of phosphatase and tensin homolog (PTEN), which is one of the major intrinsic impediments to axonal growth, aiming at improving the regenerative ability of neurons after SCI [117]. Moreover, Huang and colleagues demonstrated that siRs targeting the connective tissue growth factor (CTGF) encapsulated in MSC-EVs also has a positive effect on functional recovery after SCI [111].

Synthetic miR mimics have also been exogenously encapsulated into MSC-EVs. For instance, MSC-EVs were electroporated with the miR-132 that targets RASA1, an essential negative regulator of vascular sprouting and vessel branching. The bioengineered EVs promoted angiogenesis in vitro and enhanced neovascularization and preserved heart functions in an in vivo myocardial infarction (MI) mouse model [108]. In a study by Jing et al., MSC-EVs harbouring miR-499a-5p inhibited endometrial tumour growth and angiogenesis in vitro and in vivo, by directly targeting to upregulated gene VAV3 [109]. Using a different strategy, Wang and collaborators exogenously loaded EVs secreted by human WJ-derived MSCs (MSC(WJ)) with a miR mimic using a commercial transfection reagent. miR-326-carrying MSC-EVs suppressed the activation of the NF-κB signalling pathway and the reduced expression levels of neddylation-related enzyme molecules, inhibiting the synthesis and production of related inflammatory factors and relieving dextran sulfate sodium (DSS)-induced inflammatory bowel disease (IBD) in a mouse model, compared to unmodified MSC-EVs [118].

Protein loading into EVs was also investigated by the genetic manipulation of parental MSCs. For example, serine/threonine kinase Akt, which plays an important role in promoting cell proliferation and inhibiting cell apoptosis, was transduced into human MSC(WJ) using an adenovirus system. Western blot semi-quantification revealed that the produced EVs harboured significantly higher levels of Akt than the control EVs. The produced EVs harboured higher levels of Akt and demonstrated increased angiogenic effects in vitro and in vivo and promoted superior cardiac regeneration in an acute MI mouse model, compared to control EVs [119]. Similarly, angiopoietin-2 (Ang-2) loaded into MSC(WJ)-derived EVs through its lentiviral-based overexpression by parental cells. Ang-2-carrying EVs enhanced angiogenesis and accelerated cutaneous wound healing in vivo [120]. Moreover, EVs secreted by pigment epithelium-derived factor (PEDF)-overexpressing MSC(AT), were shown to ameliorate cerebral ischemia–reperfusion injury in an in vivo rat model by activating autophagy and suppressing neuronal apoptosis [122]. Furthermore, MSC(M) were transduced to overexpress osteoactivin. The produced MSC-EVs stimulated the proliferation and osteogenic differentiation of MSC(M) via the activation of Wnt/β-catenin signalling and promoted bone regeneration in an ovariectomized rat model of postmenopausal osteoporosis (OP) [121]. Using exogenous loading, Rajendran and colleagues encapsulated tyrosine kinase inhibitor (TKI) into EVs produced by human MSC(AT) by direct incubation or sonication. Sonicated TKI-EVs enhanced iodine avidity in radioactive iodine-refractory thyroid cancer compared with free-TKI treatment [133].

MSC-EVs have proven to be efficient delivery vehicles for small anti-cancer drugs. For example, MSCs incubated with PTX have been shown to secrete EVs presenting a high drug concentration as quantified by high-performance liquid chromatography (HPLC) analysis. PTX-loaded EVs and induced a dose-dependent inhibition of human pancreatic adenocarcinoma cell proliferation, reducing tumour growth by up to 50% [62]. Using an alternative approach, Kalimuthu et al. directly incorporated PTX into MSC-EVs by serial extrusion through 10-, 5- and 1-μm polycarbonate membrane filters. These vesicles demonstrated their significant therapeutic effects against breast cancer both in vitro and in vivo [130]. As a prospective approach to surpassing chemoresistance of the pancreatic ductal adenocarcinoma (PDAC), a combination therapy of gemcitabine monophosphate (GEMP) and PTX delivered by MSC-EVs was developed, using electroporation and sonication as loading methods, respectively [129]. Despite the low encapsulation efficiencies determined by HPLC (5.92% and 2.62% for GEMP and PTX, respectively), GEMP/PTX-loaded EVs showed a great anti-tumour efficacy in vitro and in vivo in a PDAC orthotopic mouse model [129]. Furthermore, the anti-cancer drug DOX was also successfully packed into MSC-EVs using different endogenous loading methods, including electroporation or dialysis [126,127]. UV–vis-spectroscopy-mediated quantification showed that electroporation yielded a higher DOX encapsulation efficiency with a maximum of 35% [127]. Other small anti-cancer drugs have been packed into MSC-EVs and exhibited improved therapeutic effects, including taxol (TXL) [124] and norcantharidin (NCTD) [128]. Another promising approach consists in loading other small molecule drugs into MSC-EVs to treat inflammation or tissue regeneration besides malignant tumours. For example, isolated MSC(M)-derived EVs were incubated with Cur to engineer EVs with anti-inflammatory properties. After administration into a mouse model of ischemic stroke, Cur-carrying EVs suppressed the inflammatory response and cellular apoptosis in the lesion region of an ischemic stroke mouse model more effectively than non-modified EVs or Cur alone [132].

Finally, MSC-EVs can be packed with synthetic NPs. For instance, magnetic NPs were incorporated into MSC-EVs, using an MSC-mediated assembly process. Essentially, MSCs were incubated with iron oxide NPs and the secreted EVs were loaded with the NPs. After injection and magnet guidance, the NP-harbouring EVs showed significantly enhanced accumulation at the site of injured skin, demonstrating a capacity to induce faster wound reduction with increased collagen deposition and high blood vessel density [123]. Similarly, in a study by Altanerova et al., MSC-EVs were loaded with Venofer, carbohydrate-coated ultrasmall superparamagnetic iron oxide nanoparticles (SPIONs), by incubating MSCs with a Venofer–heparin–protamine sulphate complex overnight. The secreted Venofer-carrying EVs were successfully internalized by the tumour cells and facilitated their ablation via cytotoxic hyperthermia by applying an alternating magnetic field [125]. Some studies have also reported the modification of cargo of MSC-EVs by hybrid membrane engineering strategies. Singh and colleagues assembled MSC(M)-derived EVs and liposomes containing polypyrrole (Ppy) NPs, using the freeze–thaw method. Ppy-NPs naturally possess electrical conductivity, which can promote nerve regeneration and ameliorate diabetic peripheral neuropathy (DPN). After intramuscular injection into a DPN mouse model, Ppy-NP-encapsulating hybrids in combination with electrical stimulation reduced the neural and muscular damage [131].

### 4.2. Improving the Therapeutic Potential of MSC-EVs via Surface Engineering

Apart from cargo modification, different bioengineering strategies have been used on MSC-EVs to functionalize their surface. Essentially, by genetic engineering of parental MSCs or direct chemical modification of isolated EVs, the surface of MSC-EVs has been manipulated to enhance their therapeutic properties and improve target selectivity, aiming to develop potent targeted therapies with reduced adverse effects (Table 2).

The conjugation of peptides on the surface of MSC-EVs has been shown to improve their targeting towards particular organs or tissues, demonstrating prospective effects in the treatment of different types of cancer, heart and brain diseases. Many researchers have been using genetic engineering to generate recombinant peptides that are displayed on the surface of MSC-EVs, usually by fusing a targeting ligand to an EV membrane-enriched peptide/protein. Envisioning the targeted delivery of drugs to the brain, Yang and colleagues developed neuron-specific targeting EVs by engineering MSCs to overexpress Lamp2b fused with RVG. After systemic administration into a mouse model of cortical ischemia, RVG-displaying MSC-EVs efficiently deliver the exogenously loaded miR-124 to the ischemic region and ameliorate brain injury by promoting neurogenesis [107]. Similarly, MSC(M) were transduced to overexpress Lamp2b fused with ischemic myocardium-targeting peptide (IMTP) CSTSMLKAC and produce cardiac-cell-targeting EVs. Intravenously injected IMTP-displaying EVs showed enhanced accumulation in the MI region and significantly increased capillary density, inhibited inflammatory response, reduced infarct size and preserved cardiac function, compared to naked EVs [137]. Alternatively, in a study by Wang et al., a peptide targeting cardiac troponin I (cTnI), which is highly expressed in the MI, was used as an EV membrane-displaying ligand for the targeted delivery of miR-590-3p to the ischemic area. The MSC-EVs decorated with the cTnI-targeting peptide effectively accumulated in the infarct area along the cTnI concentration gradient [110]. Gomari and collaborators improved the efficiency of MSC-EVs for targeted anti-cancer drug delivery by transducing the parental cells with a lentivirus encoding Lamp2b fused with HER2-specific designed ankyrin repeat protein (DARPin), which are synthetic peptides with high binding affinity and specificity to their target protein. The engineered EVs were preferentially uptaken by HER2-overexpressing breast cancer cells compared to normal cells, effectively delivering DOX and siR molecules [134].

Surface modification of MSC-EVs can be used not only to improve targeting but to introduce an additional therapeutic moiety. For instance, tumour necrosis factor (TNF-α)-related apoptosis-inducing ligand (TRAIL) is a widely studied anti-cancer agent that selectively triggers an extrinsic apoptotic pathway in malignant cells [149]. In this context, Yuan and colleagues found that EVs secreted by genetically engineered TRAIL-expressing MSCs selectively induced apoptosis in eleven cancer cell lines and were able to partially overcome TRAIL resistance in cancer cells [140]. In a study by Zhang et al., MSCs were transfected to overexpress a plasmid encoding fusion protein of cell-penetrating peptides (CPP) and TNF-α which resulted in the secretion of EVs with TNF-α anchored in the membrane. Compared to unmodified EVs, TNF-α-EVs significantly enhanced tumour cell growth inhibition through induction of the TNFR-I-mediated apoptotic pathway in vitro and in vivo [139]. Xu et al. proposed a platform for the treatment of autoimmune disease by developing activated immune-cell-specific targeting EVs. For that, MSCs were modified to overexpress programmed cell death ligand 1 (PD-L1), whose receptor is highly expressed in autoimmune pathological tissues and involved in the signalling pathway of inhibition of immune responses and preservation of immune homeostasis. The PD-L1-expressing MSC-EVs were recognized by various activated immune cells including T cells, macrophages and dendritic cells with high expression of PD-L1 receptor, in a DSS-induced colitis mouse model. Additionally, the engineered EVs restored tissue lesions by reconfiguring the local immune microenvironment [138]. Moreover, MSC(AT)s were engineered with lentivirus encoding interleukin 2 (IL-2), a cytokine that stimulates anti-cancer immunity, for its EV-mediated delivery, aiming to reduce systemic toxicity. IL-2-EVs were able to activate human CD8^+^ cytotoxic T cells, which effectively killed human triple-negative breast cancer cells; however, these failed to suppress the proliferation of human peripheral blood mononuclear cells (PBMCs) [135]. In a study by Conceição et al., MSCs were engineered to produce EVs displaying pro-inflammatory cytokine interleukin 6 signal transducer (IL-6ST) decoy receptors at their surface to selectively inhibit the IL-6 trans-signalling pathway, a specific mediator in chronic inflammatory responses, while not interfering with the classical signalling properties of this cytokine. IL6ST decoy receptor-decorated EVs demonstrated their decoy activity by inducing a reduction in STAT3 phosphorylation in the quadriceps and gastrocnemius muscles of a Duchenne muscular dystrophy mouse model [136].

Chemical engineering has been widely investigated in incorporating targeting moieties into the surface of MSC-EVs, including peptides, RNA/DNA aptamers and drugs. For example, Zhang et al. conjugated the surface of MSC-EVs with the c(RGDyK) peptide, known to target the ischemic brain by binding to integrin α_v_β_3_ in reactive cerebral vascular endothelial cells, using bio-orthogonal copper-free click chemistry. Essentially, the reactive dibenzylcyclootyne-conjugated EVs formed a covalent bond with an azide group on the lysine of the c(RGDyK) peptide. After intravenous administration into a mouse model of ischemic stroke, the engineered EVs successfully targeted lesions within ischemic brain tissue [132]. This strategy allowed the accumulation of EV-loaded cholesterol-modified miR-210 in the lesion region and promoted microvascular angiogenesis [116]. In another study, MSC-EVs were chemically functionalized via a reaction between an aptamer-conjugated aldehyde and the amino group of EV-membrane proteins. Basically, the surface of EVs was conjugated with an MSC(M)-specific RNA aptamer to improve BM targeting. After intravenous injection, the engineered EVs successfully targeted the BM and promoted bone regeneration in OP and femur fracture mouse models, in contrast to non-functionalized EVs, which accumulated in the liver and lungs [141]. Similarly, Bagheri and colleagues engineered the surface of MSC(M)-derived EVs with the 5TR1 DNA aptamer that has a high affinity with MUC1, a transmembrane mucin glycoprotein overexpressed in different types of cancer cells. Click chemistry led to the formation of a covalent bond between carboxylate-modified 5TR1 aptamer and the amine group on the surface of EVs. After intravenous injection into a mouse model of colon adenocarcinoma, the 5TR1-aptamer-EVs exhibited higher tumour accumulation and faster liver clearance in comparison with unmodified EVs [127]. Using the same reaction, Shamili et al. conjugated MSC-EVs with the LJM-3064 DNA aptamer which has a strong affinity toward myelin, and demonstrated remyelination induction, aiming to establish a novel approach for managing multiple sclerosis (MS). LJM-3064-aptamer-EVs showed a higher affinity for the myelin basic protein-producing cells in vitro, and synergistically induced immunomodulatory and remyelination effects in the experimental mouse model of MS [143]. To overcome the immunosuppressive tumour microenvironment of PDAC, an EV-based dual DDS of siR-galectin-9 was developed to block the galectin-9/dectin-1 axis and reverse immunosuppression caused by tumour-associated macrophages, and prodrug oxaliplatin (OXA), to act as immunogenic cell death trigger and kill the tumour cells by inhibiting DNA synthesis and repair. After exogenous loading of siR-galectin-9, OXA was added to the MSC-EVs obtaining a stable maleimide–thiol conjugate through vortexing [112]. MSC-EVs cancer-targeted delivery can also be achieved with magnetism. For example, SPIONs were conjugated with transferrin (Tf) using click chemistry. Afterwards, Tf-SPIONs were assembled to the surface of MSC-EVs by transferrin–transferrin receptor-mediated interaction. The engineered EVs were used for cancer-targeted delivery of TNF-α (described above), under an external magnetic field in a mouse model of melanoma subcutaneous cancer [139].

Another chemical strategy used to engineer the surface of MSC-EVs has been lipid assembly. For instance, Gangadaran and colleagues functionalized the surface of MSC-EVs with a peptide that targets interleukin-4 receptor (IL-4R), which is upregulated in various types of tumours, using a membrane phospholipid-based linker composed of dioleylphosphatidylethanolamine (DOPE), methoxy PEG and succinyl-N-hydroxy-succinimidyl (NHS) ester. The IL-4R-targeting peptide EVs induced a faster internalization into human anaplastic thyroid cancer cells in vitro compared to EV displaying a control peptide. Additionally, engineered EVs were shown to efficiently target tumours in a xenograft mouse model, in contrast to control EVs that are predominantly localized in the liver and spleen [142]. Using a similar strategy, MSC(M)-derived EVs were conjugated with the RVG peptide using a DOPE-NHS linker. The RVG-displaying EVs enhanced their binding to the cortex and hippocampus upon intravenous administration in a mouse model of AD, ameliorating spatial learning and memory impairments [144]. In a study by Cui et al., bone-targeting EVs were developed through conjugation with the peptide SDSSD modified with a diacyl lipid tail via hydrophobic insertion. The peptide-displaying EVs specifically delivered the exogenously loaded siR targeting schnurri-3 (SHN3) to osteoblasts and bone-forming surfaces via SDSSD/periostin interactions [115].

Feng and colleagues engineered MSC-EVs with a positively charged surface by simple incubation with a novel cationic amphiphilic macromolecule ε-polylysine (εPL)-PEG-DSPE, in order to enhance EVs intra-articular bioavailability in OA therapy. In contrast with unmodified EVs, electropositive MSC-EVs demonstrated increased chondrocyte uptake and retention ability in cartilage, leading to an enhanced OA treatment [145].

Some studies have also modified the surface of MSC-EVs by engineering hybrid nanocarriers. For instance, the PEG-mediated fusion of MSC-EVs with functionalized liposomes with various liposome-to-EV ratios has allowed the manipulation of the EV membrane properties, namely cellular uptake. In fact, PEGylated liposome–EV hybrids enabled a lower internalization by macrophages in situ [84]. In another study, membrane MSC-EVs were fused with platelet membrane fractions in the presence of PEG, in order to enhance their accumulation in injured tissues. Compared to unmodified MSC-EVs, the cellular uptake of hybrid EVs was significantly enhanced in endothelial cells and cardiomyocytes, but not macrophages. Additionally, the hybrid EVs showed improved targeting to injured myocardium and enhanced therapeutic potency in a mouse model of MI [148]. Similarly, Zhang and colleagues generated monocyte-mimic–EV hybrids to improve the delivery efficiency of MSC-EVs to ischemic myocardium, by mimicking the recruitment feature of monocytes [147]. Moreover, Lee and collaborators fabricated EV hybrids by fusing the membrane of MSC(WJ) and macrophages through the serial extrusion of cells via microporous and nanoporous filters. The engineered hybrid EVs largely accumulated in the SCI area after the in vivo systemic injection, due to the increased levels of ischemic-region-targeting molecules compared to MSC-EVs [146].

## 5. Clinical Translation of Bioengineered MSC-EVs

In the past few years, more than thirty clinical trials have been registered to address the innate potential of MSC-EVs for the treatment of different diseases, including bronchopulmonary dysplasia (NCT03857841), burn wounds (NCT05078385), OA (NCT05060107), AD (NCT04388982), dystrophic epidermolysis bullosa (NCT04173650), CD (NCT05130983), periodontitis (NCT04270006) and COVID-19-associated pneumonia (NCT04276987; NCT04491240) (listed in “clinicaltrials.gov” on 1 February 2023 using the terms “MSC exosomes OR MSC extracellular vesicles”). However, there are still several hurdles hampering the clinical application of non-modified MSC-EVs, namely their short half-life, poor targeting ability, rapid clearance from the target area and inefficient payload [69]. In order to surpass these limitations, bioengineering strategies have been implemented in MSC-EVs and have demonstrated great results in pre-clinical animal models, as herein described (Table 1 and Table 2).

Clinical studies have already been exploring MSC-EVs as DDS of nucleic acids for the treatment of different diseases. miR-124-loaded MSC-EVs have been found to ameliorate brain injury by promoting neurogenesis after ischemia [107]. In this context, a phase I/II clinical trial (NCT03384433) is evaluating the effect of allogenic MSC-EVs enriched with miR-124 as a treatment for acute ischemic stroke patients. The patients are expected to receive the miR-124-MSC-EVs by intraparenchymal injection, one month after stroke onset. In this study, measurements of treatment-derived adverse events, including stroke recurrences, brain oedema and seizures and measurements of the degree of disability of stroke patients will be conducted within a period of 12 months after therapy administration. Additionally, MSC-EVs containing siR targeting oncogenic Kras^G12D^ mutations are being tested against PDAC in a phase I clinical trial (NCT03608631). PDAC patients are expected to receive the siR-KRAS^G12D^-EVs through intravenous administration on days 1, 4 and 10, with repeated treatments every 14 days. The primary objectives of this study are the assessment of a maximum tolerated dose (MTD) and the identification of dose-limiting toxicities (DLT). Secondary objectives include the pharmacokinetics of circulating EVs, the assessment of overall response and disease control rates and the evaluation of the median progression-free survival and median overall survival with therapy. Another phase I clinical trial (NCT05043181) will be testing the therapeutic potential of low-density lipoprotein (LDL) receptor (LDLR)-mRNA in the treatment of homozygous familial hypercholesterolemia (HoFH). HoFH patients carry a functional loss mutation of the LDLR gene causing severely elevated plasma LDL cholesterol and premature coronary heart disease [150]. In this clinical study, MSC(M) will be engineered with an LDLR-expressing virus vector and the produced LDLR-mRNA-enriched EVs will be used as HoFH therapy. A total of three treatments with an interval of 7 ± 1 days will be injected into the patients through an abdominal puncture, testing six EV doses. The primary outcome will be measuring the changes in total cholesterol, LDL cholesterol, high-density lipoprotein cholesterol and triglyceride, and the secondary outcome will be the assessment of the degree of coronary stenosis and the volume and stability of carotid artery plaques.

Despite the substantial amount of research on bioengineered MSC-EVs as improved drug delivery therapeutics, only a few of them have been investigated in clinical settings. One major contributing factor is that most methods used in pre-clinical models for the production and isolation of EVs have low yields, insufficient purity profiles and are hardly scalable (e.g., conventional planar culture systems such as T-flasks, ultracentrifugation and precipitation-based isolation methods) [8]. In an attempt to overcome these limitations, Haraszti and colleagues developed a robust and scalable strategy to produce and isolate EVs from MSC(WJ) compatible with good manufacturing practices (GMPs). Through the combination of scalable microcarrier-based 3D cultures and tangential flow filtration, the yield of EVs was increased by 140-fold in comparison to 2D cultures coupled with ultracentrifugation. Interestingly, these EVs were seven-fold more active in their ability to transfer therapeutic siR to primary neurons compared to EVs produced in 2D cultures and isolated by ultracentrifugation [151]. An alternative strategy to increase EV yields is through genetic modification of the parental cells. For instance, human MSC(M) were engineered to overexpress metalloreductase STEAP3, syndecan-4 and L-aspartate oxidase proteins, which are involved in the biogenesis of Exo, significantly increasing EV production [152]. Another approach to overcome the challenge of scalability is using cell-derived nanovesicles (CDNs) that are EV mimics formed by the serial extrusion of cells through filters. CDNs generation strongly reduces production time and cost, while potentially increasing production yield by up to 250-fold [153]. Wang and collaborators demonstrated that the yield of extruded MSC-derived CDNs was 20-fold higher than that of secreted EVs, and the myocardial protective effects in a MI mouse model were maintained [154].

Furthermore, the reproducible manufacture of an EV-based product at a clinical scale is challenging when using MSCs as parental cells due to their limited lifespan and inherent batch-to-batch or donor-to-donor variations [155]. MSC immortalization is a possible approach to tackle these limitations and facilitate large-scale EV production. Some studies have reported that immortalization (e.g., by *MYC* transgene integration) did not confer tumorigenic activity to MSC and their secreted EVs [156]. Still, MSC-EVs produced by immortalized cells will always raise safety concerns in what concerns their tumour-promoting effects. An alternative approach to increase the yield and homogeneity of MSC-EVs is using induced pluripotent stem cells (iPSC) as a source of MSCs [115,145]. iPSC-derived MSCs potentially allow unlimited cell supply, which lowers manufacturing costs and increases the scalability potential for the production of a GMP-grade EV product [155].

Importantly, most of the methods reported to bioengineer the content and surface of MSC-EVs are still at the pre-clinical level and with limited scalability. Due to the lower loading efficiencies often associated with exogenous loading, cargo engineering of MSC-EVs has mostly been explored by endogenous loading through the genetic modification of the parental cells, which is still troublesome in primary cells and has unpredictable loading efficiencies [59,60]. In order to address these limitations, Yang et al. established a novel cellular-nanoporation-based strategy for the large-scale loading of mRNA into MSC-EVs. Essentially, MSCs were cultured on a specifically designed biochip, transfected with plasmid DNAs and induced to release EVs harbouring the transcribed RNAs by an electrical stimulus. Compared with bulk electroporation, cellular nanoporation generated up to 50-fold more EVs with 1000-fold higher levels of transcripts [157]. Furthermore, surface modification of previously isolated MSC-EVs has shown great signs of progress regarding efficiency and scalability, avoiding the complexity of genetic engineering strategies [59].

Overall, apart from the innate heterogeneity of MSCs and the respective secreted EV populations, the differences among the techniques available for the production, isolation, characterization and modification of EVs hinder the cross-comparison between different studies and thus the evaluation of subsequent progress of the field. Thus, a simple, cost-effective and streamlined manufacturing process for bioengineered MSC-EVs is needed to facilitate their clinical translation [8,60].

## 6. Conclusions

EVs have undergone a significant shift in perception in the last decade, emerging from promising diagnostic biomarkers, and now being considered as promising therapeutic agents with intrinsic regenerative properties, as well as prospective nanocarriers for drug delivery with enhanced biocompatibility and inherent targeting capabilities. Due to their safety profile and innate therapeutic properties, MSCs have been depicted as an excellent EV-producing cell line. Significant advances have been made in developing strategies to bioengineer the MSC-EVs to further increase their circulation half-life, targeting and accumulation to disease sites, and efficiently deliver desired therapeutic molecules. The functionalization of MSC-EVs with targeting ligands using genetic manipulation or chemical modification makes them more directed and efficient therapeutics. Moreover, the modification of the intraluminal composition of MSC-EVs through their complementation with specific exogenous payloads potentially enables the establishment of customized treatments. In this context, the combination of these two bioengineering strategies is expected to contribute to the development of personalized MSC-EV-based therapies with improved targeting and therapeutic potency in the treatment of a multitude of diseases, including cancer, brain and heart disorders, as well as bone injuries (Figure 3). Nevertheless, the clinical translation of EV-based therapeutics is still humped by the lack of standardized and robust methods for the production, isolation and characterization of EVs. On the other hand, MSC-EV-based drug delivery benefits are highly dependent on the nature of the therapeutic payload, the loading method, the targeted disease site and the mechanism of action. Therefore, the establishment of efficient and reproducible methods to engineer the targeting and drug loading of EVs is greatly needed and will undoubtedly remain a major focus of future research in the field.

## Figures and Tables

**Figure 1 biomedicines-11-01231-f001:**
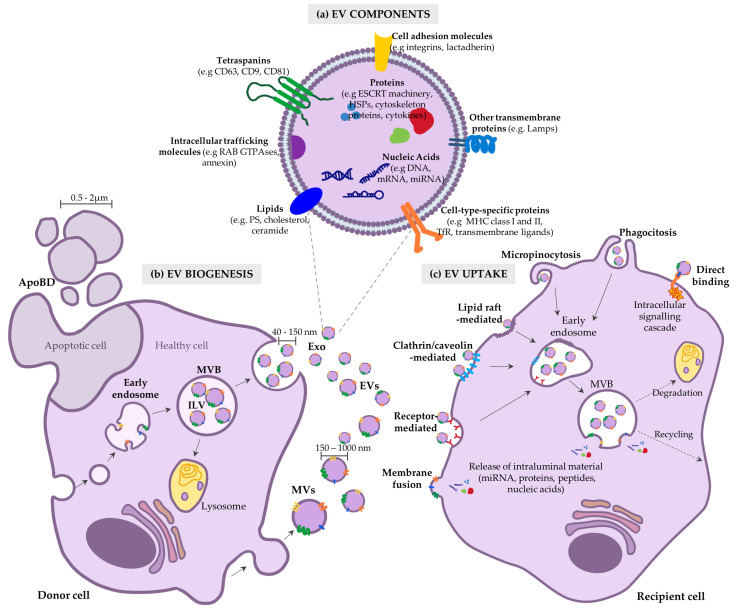
Basics of extracellular vesicle (EV) biology. (**a**) General composition of EVs: EVs are nano-sized lipid bilayer structures that enclose a variety of cellular components including cytosolic and transmembrane proteins, bioactive lipids and nucleic acids. (**b**) Biogenesis of the different subsets of EVs: EVs are formed either by the disassembly of an apoptotic cell into subcellular fragments as apoptotic bodies (ApoBD), the budding of the plasma membrane, in which case they are referred to as microvesicles (MVs) or as intraluminal vesicles (ILVs) within the lumen of multivesicular bodies (MVBs). MVBs fuse with the plasma membrane to release ILVs that are then called exosomes (Exo). (**c**) Mechanisms of uptake of EVs by the recipient cells: EVs can induce a downstream signalling cascade in the recipient cell via direct binding or transfer of their intraluminal content by membrane fusion or endocytosis-mediated internalization. The internalized EVs follow the endosomal pathway and can either be recycled back to the plasma membrane, degraded in the lysosome or undergo endosomal escape releasing their intraluminal cargo. ESCRT—endosomal sorting complex required for transport; HSP—heat shock protein; Lamp—lysosomal-associated membrane proteins; MHC—major histocompatibility complex; PS—phosphatidylserine; TfR—transferrin receptor.

**Figure 2 biomedicines-11-01231-f002:**
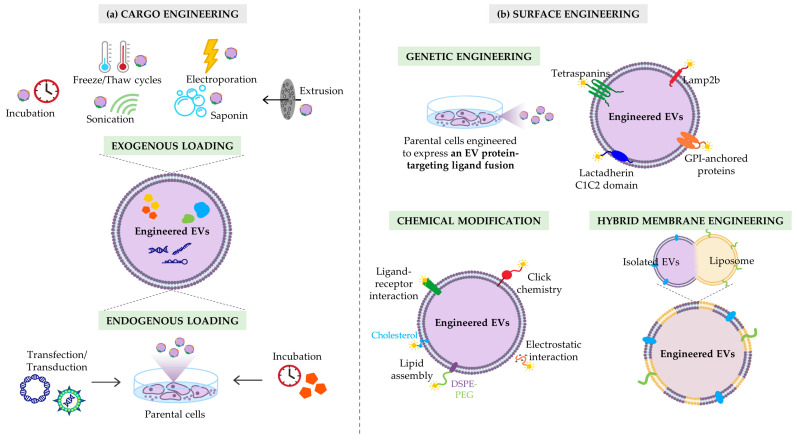
Strategies to maximize the therapeutic efficacy of extracellular vesicles (EVs). (**a**) Cargo engineering of EVs by exogenous/direct loading, through external incorporation of cargo into isolated EVs, or endogenous/indirect loading, by providing the parental cells with the means to naturally incorporate the desired cargo during EV biogenesis. (**b**) Surface engineering of EVs: The parental cells can be genetically engineered to produce EVs displaying transmembrane protein-targeting ligand fusions. The isolated EVs can be chemically modified, by anchoring targeting moieties to the surface of isolated EVs through covalent bonds, lipid self-assembly or other non-covalent reactions. Hybrid membrane engineering allows the fusion of isolated natural EVs and synthetic liposome nanoparticles. DSPE—1,2-distearoyl-sn-glycero-3-phosphoethanolamine; GPI—glycosyl-phosphatidylinositol; Lamp—lysosomal-associated membrane proteins; PEG—polyethylene glycol.

**Figure 3 biomedicines-11-01231-f003:**
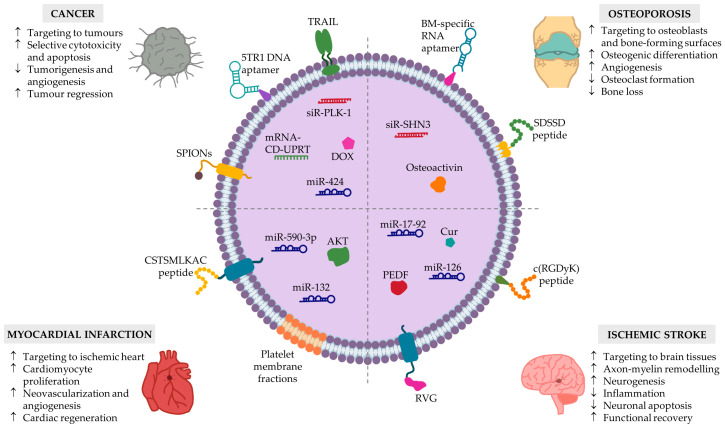
Representative applications of bioengineered MSC-EVs with improved therapeutic payload and target specificity in the treatment of cancer [93,98,113,115,127,140,141], osteoporosis [116,122,142], myocardial infarction [109,111,120,138,149] and ischemic stroke [96,99,108,117,123,133]. CD-UPRT—cytosine deaminase fused to uracil phosphoribosyltransferase; Cur—curcumin; DOX—doxorubicin; PEDF—pigment-epithelium-derived factor; PLK-1—serine/threonine protein kinase; RVG—rabies viral glycoprotein; SHN3—schnurri-3 protein; SPIONs—superparamagnetic iron oxide nanoparticles; TRAIL—tumour necrosis factor-related apoptosis-inducing ligand.

**Table 2 biomedicines-11-01231-t002:** Overview of the potential strategies and applications of surface-engineered mesenchymal-stromal-cell-derived extracellular vesicles (MSC-EVs).

Type of Strategy	Surface Modification	Application	Therapeutic Effect	MSC Source	Ref.
Genetic surface engineering	cTnI-targeting peptide	Myocardial infarction	Improved targeting to ischemic heart	Rat BM	[110]
	HER2-specific DARPins	Breast cancer	Improved uptake by HER2-positive cells	N/A	[134]
	IL-2	Cancer	Activated human CD8^+^ T-killers	Human AT	[135]
	IL-6ST decoy receptors	Duchenne muscular dystrophy	Counteracted the effects of pathological signalling pathways	Human BM	[136]
	CSTSMLKAC peptide	Myocardial infarction	Improved targeting to ischemic heart	Mouse BM	[137]
	PD-L1	Autoimmune Diseases	Improved recognition and inactivation of immune cells	Mouse BM	[138]
	RVG	Ischemic stroke	Increased targeting to ischemic brain	Mouse BM	[107]
	TNF-α	Cancer	Inhibited tumour growth	Human N/A	[139]
	TRAIL	Cancer	Increased selective apoptosis	Human N/A	[140]
Chemical surface engineering	5TR1 DNA aptamer	Colon adenocarcinoma	Improved targeting to tumours	Mouse BM	[127]
	BM-specific RNA aptamer	Osteoporosis	Improved targeting to bone marrow	Mouse BM	[141]
	c(RDGyK) peptide	Ischemic stroke	Improved targeting to ischemic brain	Mouse BM	[116,132]
	IL-4R-targeting peptide	Anaplastic thyroid cancer	Improved targeting to tumours	Human BM	[142]
	LJM-3064 aptamer	Multiple sclerosis	Increased affinity to myelin-producing cells; induced immunomodulatory and remyelination effects	Mouse BM	[143]
	OXA	Pancreatic ductal adenocarcinoma	Induced immunogenic tumour cell death	Human BM	[112]
	RVG	Alzheimer’s disease	Improved targeting to brain tissues	Mouse BM	[144]
	SDSSD peptide	Osteoporosis	Improved targeting to osteoblasts and bone-forming surfaces	iPSC	[115]
	SPION	Melanoma subcutaneous cancer	Improved targeting under an external magnetic field	Human N/A	[139]
	εPL-PEG-DSPE	Osteoarthritis	Increased uptake and retention in cartilage	iPSC	[145]
	Macrophage membranes fractions	Spinal cord injury	Increased levels of ischemic region-targeting molecules and improved targeting to injury	Human WJ	[146]
Hybrid membrane engineering	Monocyte membranes fractions	Myocardial infarction	Improved targeting to ischemic myocardium	Rat BM	[147]
	PEGylated liposomes	Cancer	Decreased internalization by macrophages	Mouse BM	[84]
	Platelet membrane fractions	Myocardial infarction	Improved targeting to injured myocardium and enhanced cellular uptake by endothelial cells and cardiomyocytes	Human BM	[148]

AT—adipose tissue; BM—bone marrow; cTnI—cardiac troponin I; DARPin—designed ankyrin repeat protein; DSPE—1,2-distearoyl-sn-glycero-3-phosphoethanolamine; HER2—human epidermal growth factor receptor 2; IL-2—interleukin 2; IL-4R—interleukin-4 receptor; IL-6ST—cytokine interleukin 6 signal transducer; iPSC—induced pluripotent stem cells; OXA—oxaliplatin; PD-L1—programmed cell death-ligand 1; PEG—polyethylene glycol; RVG—rabies viral glycoprotein; SPION—superparamagnetic iron oxide nanoparticles; TNF-α—tumour necrosis factor; TRAIL—tumour necrosis factor-related apoptosis-inducing ligand; WJ—Wharton’s jelly; εPL—ε-polylysine.

## Data Availability

Not applicable.

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
