# Peer review of "Bioengineered Mesenchymal-Stromal-Cell-Derived Extracellular Vesicles as an Improved Drug Delivery System: Methods and Applications"

_biomedicines, 2023, doi:10.3390/biomedicines11041231_

Round 1

Reviewer 1 Report

This review article is a very good one that deals with reviewing the use of Bioengineered Mesenchymal Stromal Cells-derived Extracellular Vesicles as an Improved Drug Delivery System.

This review article is a very good and a novel one that deals with reviewing this hot topic.
Extracellular vesicles seem to be internalized more efficiently and deliver their therapeutic agent several orders of magnitude more efficiently than synthetic nanoparticles. 

There is a paucity in articles reviewing this new trend of exploiting the mesenchymal cells in drug delivery so this article helps in filling this gap.

It would just be more beneficial for the readers to include a separate section about the use of MSCs in regenerative medicine and tissue engineering such as in skin burns and so on.

The topic is new and is well addressed and the figures and tables are informative and comprehensive.

The references are appropriate in number and in relevancy.

Also providing the readers some information about the possible ways of detecting the cargo loading efficiency (especially with the use of small molecular weight drugs) in MSCs would be a great asset to the manuscript.

Author Response

The authors would like to acknowledge the reviewers her/his constructive feedback and suggested revisions due to which the manuscript has been improved. We have taken these comments together to revise our manuscript. We provide a point-by-point response to her/his comments below. The changes added to the revised manuscript are highlighted in red.

Comment 1: “It would just be more beneficial for the readers to include a separate section about the use of MSCs in regenerative medicine and tissue engineering such as in skin burns and so on.”

The authors appreciate the reviewer's suggestion and have added the following information to the introduction of the revised version of the manuscript:

  • Line 46: “For instance, MSC significantly support wound healing by polarizing macrophages anti-inflammatory M2 activation, promoting angiogenesis and enhancing the survival and migration of fibroblasts [16]. In the context of central nervous system (CNS) regeneration, MSC can facilitate neurogenesis by preventing apoptosis of endogenous neural cells and promoting axon re-extension by inhibiting the effect of extrinsic factors derived from the external environment of damaged areas [17].”

  1. Guillamat-Prats, R. The Role of MSC in Wound Healing, Scarring and Regeneration. Cells 2021, 10, doi:10.3390/cells10071729.
  2. Li, M.; Chen, H.; Zhu, M. Mesenchymal Stem Cells for Regenerative Medicine in Central Nervous System. Front Neurosci 2022, 16, 2147, doi:10.3389/FNINS.2022.1068114/BIBTEX.

Comment 2: “Also providing the readers some information about the possible ways of detecting the cargo loading efficiency (especially with the use of small molecular weight drugs) in MSCs would be a great asset to the manuscript.”

Regarding the topic of detecting the cargo loading efficiency in MSC-EVs, the authors have added the following sentences to the revised version of the manuscript:

  • miR detecting method
    • Line 429: “Quantitative PCR demonstrated that the produced EVs contained approximately 60-fold higher levels of miR-124a, compared to non-modified MSC-derived EVs.”

  • Protein detecting method
    • Line 528: “Western blot semi-quantification revealed that the produced EVs harboured significantly higher levels of Akt than the control EVs.”
  • Small drugs detecting method
    • Line 546: “For example, MSC incubated with PTX have been shown to secrete EVs presenting a highdrug concentration as quantified by high-performance performance liquid chromatography (HPLC) analysis.”
    • Line 556: “Despite the low encapsulation efficiencies determined by HPLC (5.92% and 2.62% for GEMP and PTX, respectively), GEMP/PTX-loaded EVs showed a great anti-tumour efficacy in vitro and in vivoin a PDAC orthotopic mouse model [129].”
    • Line 561: “UV–Vis spectroscopy-mediated quantification showed that electroporation yielded a higher DOX encapsulation efficiency with a maximum of 35% [127].

Reviewer 2 Report

This review article by Ulpiano et al. entitled Bioengineered Mesenchymal Stromal Cells-derived Extracellular Vesicles as an Improved Drug Delivery System is an excellent manuscript that is easy to understand and summarizes a vast body of knowledge.
However, while there is a lot of basic information on Vesicles, the mechanism of its action in cells is not well explained and may not be well understood by the reader. Additional description is needed in this regard.

The figures seem rough on my computer screen, so they need to be corrected. Also, as shown in line 192, some descriptions are incomplete and need to be corrected, including the font.

Additional Comments: 

Yes, the paper is relevant to the field.

Conclusions are consistent with evidence presented.

The references are appropriate.

Author Response

The authors would like to acknowledge the reviewer her/his constructive feedback and suggested revisions due to which the manuscript has been improved. We have taken these comments together to revise our manuscript. We provide a point-by-point response to her/his comments below. The changes added to the revised manuscript are highlighted in red.

Comment 1: “However, while there is a lot of basic information on Vesicles, the mechanism of its action in cells is not well explained and may not be well understood by the reader. Additional description is needed in this regard.”

The authors appreciate the reviewer's suggestion and have added the following information on the mechanism of action of EVs on the recipient cells to the revised version of the manuscript:

  • Line 139: “For instance, EVs have been reported to act as carriers in the long-range transfer of the canonical lipid-anchored morphogens Hedgehog (Hh) and Wnts to recipient cells which induce several physiological processes, such as stem cell maintenance, tissue repair and metabolism [39].”
  • Line 166: “Although the mechanism by which cells discriminate the fate of the internalized EVs is poorly understood, the delivery capacity of EVs has been largely demonstrated. The release of their intraluminal content triggers alterations in the recipient cells by the action of nucleic acids, including miR and mRNA, that regulate gene expression, and other important genetic elements, including genomic DNAs, mitochondrial DNAs and long noncoding RNAs [1,7]. EVs also release protein and peptide cargos that induce a functional response in the recipient cells. For example, in dentritic cells, protein cargos of EVs can be processed and used in antigen presentation regulating immune response [1,40].”

  1. Parchure, A.; Vyas, N.; Mayor, S. Wnt and Hedgehog: Secretion of Lipid-Modified. Trends Cell Biol2018, 28, 157, doi:10.1016/J.TCB.2017.10.003.
  2. Morelli, A.E.; Larregina, A.T.; Shufesky, W.J.; Sullivan, M.L.G.; Stolz, D.B.; Papworth, G.D.; Zahorchak, A.F.; Logar, A.J.; Wang, Z.; Watkins, S.C.; et al. Endocytosis, Intracellular Sorting, and Processing of Exosomes by Dendritic Cells. Blood 2004, 104, 3257–3266, doi:10.1182/BLOOD-2004-03-0824.

Round 2

Reviewer 2 Report

Thanks for the appropriate corrections.

Author Response

Dear Reviewer,

thank you for your valuable contribution in reviewing this manuscript.

Best regards

Gabriel Monteiro